# Rheological and Mechanical Properties of Bentonite–Cement Paste Reinforced with Basalt Fibers

**DOI:** 10.3390/ma16083226

**Published:** 2023-04-19

**Authors:** Pinghe Sun, Bangdi Wei, Erneste Habiyakare, Bin Bin, Le Wang, Chunlei Peng, Wenlong Ji, Han Cao, Hanhan Yang

**Affiliations:** 1Key Laboratory of Metallogenic Prediction of Nonferrous Metals and Geological Environment Monitoring, Ministry of Education, Changsha 410083, China; pinghesun@csu.edu.cn (P.S.); 205011102@csu.edu.cn (B.W.); habiyakare900@gmail.com (E.H.);; 2School of Geosciences and Info-Physics, Central South University, Changsha 410083, China; 3Hunan Hongyu Engineering Group Co., Ltd., Changsha 410007, China; 4No. 4 Branch Company of China Petroleum Pipeline Engineering Co., Ltd., Langfang 065000, China

**Keywords:** bentonite–cement paste, basalt fiber, rheological, mechanical strength, microstructure

## Abstract

Bentonite cement paste (BCP) is among the grouting materials used widely in large-pore grouting and karst cave treatment. The mechanical properties of bentonite cement paste (BCP) will be improved by additional basalt fibers (BF). In this study, the effects of basalt fiber (BF) contents and their lengths on the rheological and mechanical properties of bentonite cement paste (BCP) have been examined. Yield stress (YS), plastic viscosity (PV), unconfined compressive strength (UCS), and splitting tensile strength (STS) were used to evaluate the rheological and mechanical properties of basalt fiber-reinforced bentonite cement paste (BFBCP). Scanning electron microscopy (SEM) and energy-dispersive x-ray spectroscopy (EDS) characterize microstructure development. The results indicate that the Bingham model can provide the rheological behavior of basalt fibers and bentonite cement paste (BFBCP). The yield stress (YS) and plastic viscosity (PV) increase as the content and length of basalt fiber (BF) increase. The effect of fiber content on yield stress (YS) and plastic viscosity (PV) is greater than that of fiber length. The addition of basalt fiber (BF) enhanced the unconfined compressive strength (UCS) and splitting tensile strength (STS) of basalt fiber-reinforced bentonite cement paste (BFBCP) at the optimum basalt fiber (BF) content of 0.6%. The optimum basalt fiber (BF) content tends to increase as curing age increases. The basalt fiber length of 9 mm is the most effective for improving unconfined compressive strength (UCS) and splitting tensile strength (STS). The large increments in unconfined compressive strength (UCS) and splitting tensile strength (STS) were 19.17% and 28.21% for the basalt fiber-reinforced bentonite cement paste (BFBCP), with a basalt fiber length of 9 mm and content of 0.6%. Scanning electron microscopy (SEM) shows that the randomly distributed basalt fiber (BF) forms a spatial network structure in basalt fiber-reinforced bentonite cement paste (BFBCP), which composes a stress system under the action of cementation. Basalt fibers (BF) used in crack generation processes slow down the flow through bridging and occur in the substrate to improve the mechanical properties of basalt fiber-reinforced bentonite cement paste (BFBCP).

## 1. Introduction

Currently, shield tunneling in karst areas is a common challenge. Geological factors, such as cavities, groundwater, and rock fragmentation in karst formations, and geological hazards, such as shield sinking, water surges, instability of tunnel faces, ground collapse, and soil liquefaction, pose issues during tunnel construction [1]. With treatment techniques, grouting is the most frequently used water plugging and reinforcement measure [2]. However, the use of cement slurry has limitations in large loopholes and high-flowing water strata [3]. For instance, pure cement slurry is easily washed away by running water due to its poor resistance to dilution [4,5]. The treatment of caves and macropores frequently uses the grouting material bentonite cement paste (BCP) [6,7]. Grouting materials containing bentonite exhibit better resistance to dispersion than cement slurry under dynamic water conditions because of the excellent water absorption and swelling properties of bentonite [8,9]. Furthermore, bentonite, as a natural, environmentally friendly, and widely available mineral, meets the current environmental requirements of China [10]. However, the mechanical properties of bentonite cement paste (BCP) also have more limitations compared to general cement-based grouting materials. For instance, under external forces, the consolidation body is prone to penetrating cracks, lacks toughness, and exhibits weak resistance to deformation [11,12,13].

Previous studies have demonstrated that the incorporation of discrete fibers reduces the early shrinkage of grouting materials, inhibits matrix cracking, and enhances the toughness and strength of the material [14,15]. Li et al. studied the strength and anti-erosion characteristics of cement slurry based on the requirements of tunnel plugging reinforcement. The results show that the polypropylene fiber can achieve erosion resistance through the slurry at a low flow rate [16]. Shi et al. proposed an alkali-resistant glass fiber (ARGF) mixed grouting material suitable for tunnel cave reinforcement, and studied its compressive strength, tensile strength, bending strength and impermeable properties under four different amounts of fiber mixing amount (0%, 0.25%, 0.5% and 1.0%). The results showed that the addition of ARGF improved the compressive and tensile strength of the grouting with an optimal performance of 0.5% [17]. Pachta et al. investigated the performance of fiber-reinforced lime-based grout in the hardened states. The results show that the fresh state performance of the grout is slightly affected by the addition of fibers, with a slight increase in porosity and a slight decrease in apparent specific gravity and in the water absorption coefficient due to capillary action. Further, the addition of fibers (mainly fibers of 6 mm length with a binder ratio of 0.3%) increased the long-term strength and the bonding capacity of the grout [18].

However, when using grouting materials, constructors must control the slurry’s rheology in addition to its curing strength to make sure that the slurry is injected into the desired strata [19]. The incorporation of fiber will greatly affect the rheology of the slurry, which is related to the water cement ratio, fiber type, fiber content, and length of the slurry [20]. When the fiber content and length exceed a certain value, the slurry faces difficulties in flow and loses its hardening properties [21]. At a certain water–cement ratio, there is an optimal fiber content and length to improve the rheological and mechanical strength of the fiber-reinforced slurry [22]. The optimum fiber content of fiber-reinforced materials is not consistent with the influence of particle composition [23]. Therefore, it is necessary to study the influence of different fiber contents on the rheological and mechanical properties of bentonite cement paste (BCP), and understand its application.

In recent years, basalt fiber (BF) has been demonstrated to be a viable alternative to other fiber materials in geotechnical engineering due to its environmentally friendly nature. As green and ecologically friendly as wood, fiber is naturally occuring basalt with primary components of SiO_2_, Al_2_O_3_, FeO, and Fe_2_O_3_ [24]. Furthermore, basalt fibers (BF) have higher tensile strengths than E-glass and polypropylene fibers [25,26]. Additionally, compared with carbon and glass, basalt fibers (BF) have better chemical resistance to acid and alkali corrosion [27]. Furthermore, the advantage of reducing production costs also makes basalt fibers superior to carbon fibers and glass fibers [28].

In this study, basalt fibers (BF) were chosen as the modified material for bentonite cement paste (BCP). The impacts of fiber content and length on the rheological and mechanical properties of basalt fiber-reinforced bentonite cement paste (BFBCP) were explored through rheology tests, unconfined compressive strength(UCS), and splitting tensile strength(STS). Moreover, SEM and EDS tests were conducted on the microstructure of BFBCP to investigate the interaction mechanism between BF, cement hydration products, and bentonite particles. The findings will provide a valuable reference for the preparation of bentonite cement grouting materials with high performance.

## 2. Materials and Method

### 2.1. Materials and Mix Design

Bentonite is a sodium-based bentonite with montmorillonite as its main component, which is produced in Changsha, Hunan province. The 200-mesh sieve margin is less than 4% and the pulp-making rate is greater than 14 m^3^/t. The grain size of bentonite and cement is shown in Figure 1.

Ordinary Portland cement (P.O.42.5) was used as the second major component for the grout. The specific surface area of cement was 358 m^2^/kg. The initial setting time was 172 min and the final setting time was 224 min. The compressive strength after 3 days and 28 days was 5.5 MPa and 27.2 MPa, respectively. The chemical composition of the bentonite and cement is listed in Table 1.

Basalt fiber is a mineral fiber with a density of 2650 kg/m^3^ and a diameter of 13 μm. Furthermore, basalt fiber has excellent mechanical properties with tensile strengths greater than 1050 MPa, a modulus of elasticity greater than 75 GPa, and good ductility and corrosion resistance. Basalt fiber can be used as an environmentally friendly reinforcing material to improve the performance of grouting materials, and their physical and mechanical parameters are shown in Table 2.

BFBCP was composed of bentonite, cement, BF, and water. To better understand the influence of various factors on the results, a mixture coding design was used. The mix design and the corresponding proportions are given in Table 3. The water–cement ratio (W/C) of BCP was 0.8. Cement and bentonite make up 40% and 60% of the mass of the solid mixture, respectively. Additionally, it is known through the preliminary exploration experiments that when the fiber content exceeds 0.8%, the BCP faces difficulties in flow and cannot meet engineering needs. Therefore, the content of basalt fiber was set to 0, 0.2%, 0.4%, 0.6% and 0.8%; the length of the fiber was 3 mm, 6 mm, 9 mm and 12 mm; and the basalt fiber admixture was defined as the ratio of the fiber volume to the total volume of the paste.

### 2.2. Sample Preparation

It is essential to ensure that fibers are dispersed into bundled monofilaments when preparing basalt fiber-reinforced bentonite cement paste (BFBCP) [29,30]. The specific mixing procedure is depicted in Figure 2. The initial step is to dry the mixture of fibers and bentonite for 3 min, then add the appropriate amount of water and stir for 5 min to obtain a bentonite base slurry. This step is the key to the dispersion of the fibers, because in the dry state, bentonite can promote the dispersion of the fibers as monofilaments. At the same time, the bentonite base slurry has high compatibility, which can facilitate the uniform distribution of fibers in the slurry. The second step is to add the cement to the bentonite base slurry and mix rapidly for 3 min to make basalt fiber-reinforced bentonite cement paste (BFBCP). The third step is to pour the basalt fiber-reinforced bentonite cement paste (BFBCP) slurry into the molds with walls of 50 mm × 100 mm and 25 mm × 50 mm and leave them to stand for 24 h before demolding. Finally, the basalt fiber-reinforced bentonite cement paste (BFBCP) specimens are cured in a maintenance box for 7 days to 28 days. Then, the relative humidity of standard curing is greater than 95%, and the temperature ranges between 20 and 30 °C.

### 2.3. Testing Methods

#### 2.3.1. Rheology Tests

Rheological parameters are often used quantitatively for assessing microscopic effects within the slurry and are sensitive to perturbations in the mixture [31]. The soft solids rheometer (Brookfield RST-R/S) and a computer-programmed constant shear stress rheometer are used to obtain rheological parameters of the basalt fiber-reinforced bentonite cement paste (BFBCP). The test was equipped with a paddle rotor size of VT-60-30 (paddle diameter of 30 mm and height of 60 mm). Figure 3 showed that the rheological test consisted of a pre-shear phase, a stabilization phase, and a data acquisition phase. The first stage is the pre-shear stage, where the shear rate increases from 0 s^−1^ to 100 s^−1^ for the 30 s^−1^. The second is the stabilization stage where the shear stops in the 30 s. Finally, the third is the data collection stage, where the shear rate increases from 0 s^−1^ to 80 s^−1^ in the 80 s and decreases from 80 s^−1^ to 0 s^−1^ in the 80 s. Furthermore, each data point was collected in 4 s to obtain 38 data points. Only the rheological data are valid and range between 10% and 100% of the required torque.

#### 2.3.2. Unconfined Compressive Strength Tests

In this study, the unconfined compressive strength of 50 mm × 100 mm paste slurry specimens was obtained by using an electronic universal material testing machine GB/T 17671 (1999). Thus, the unconfined compressive strength is a direct indicator of the grouting material, while the compressive strength is a direct indicator of the grouting material.

#### 2.3.3. Splitting Tensile Strength Tests

Among the crucial mechanical indicators to assess a grouting material’s resistance to tensile failure is its tensile strength [32]. The tensile strength of the samples resulted from the series of Brazilian splitting tests on the cured samples. The reference specifically encompass the details of the method and mechanical principles [33]. A sample was placed in the mixture horizontally. The vertical loading rate was controlled at 0.5 mm/minute during the test and ended at the damaged samples. Then, the splitting tensile strength (*σ*) was calculated using the following equation:(1)$$\sigma =\frac{2P}{\pi dl}$$
where *σ* is the tensile strength of BFBCP, MPa; *P* is the failure load, kN; *d* is the diameter of the sample, mm; *l* is the length of the sample, mm.

#### 2.3.4. SEM Tests

This study indicated that basalt fibers (BF) and the bentonite cement paste (BCP) matrix were connected with microscopic topography observed by using a field of an emission electron microscope system. The accelerating voltage was 3 keV, and the sample size was 30 mm × 30 mm × 5 mm. Ultimately, the microstructure of basalt fiber-reinforced bentonite cement paste (BFBCP) was obtained from photographs taken at different magnifications.

## 3. Results and Discussions

### 3.1. The Rheological Model

Rheology can characterize the internal structure and macroscopic properties of a material. The rheological properties of the slurry and their inherent relationship can influence the grouting design [34]. The basalt fibers (BF) affect the viscosities of the basalt fiber-reinforced bentonite cement paste (BFBCP) (Figure 4). Therefore, the shear rate of apparent viscosities in water decreases rapidly with the increase in the shear rate of 0–20 s^−1^ and then tends to be stable. This indicates that basalt fiber-reinforced bentonite cement paste (BFBCP) has undergone shear thinning.

Currently, the Bingham model, the H-B (Herchel–Bulkley) model, power-law model, and the Ostwald model are used to describe the rheological behavior of paste [35]. The shear rate and shear stress curves of basalt fiber-reinforced bentonite cement paste (BFBCP) are indicated in Figure 5. However, the rheological behavior of basalt fiber-reinforced bentonite cement paste (BFBCP) is similar to that of Bingham fluid. Thus, previous studies have considered that the fiber slurry is similar to Bingham fluid, and the modified Bingham fluid equation is proposed Equation (2).
(2)$$\tau =\tau_{0}+\eta \dot{\gamma }+c{\dot{\gamma }}^{2}$$
where $\tau_{0}$ is the yield stress, Pa; η  is the plastic viscosity, Pa·s; c is the regression constant; τ is the shear stress, Pa; $\dot{\gamma }$ is the shear rate, s^−1^. 

In the study of the effect of basalt fibers (BF) on the rheological behavior of basalt fiber-reinforced bentonite cement paste (BFBCP), the modified Bingham rheological model was used to fit the shear stress data as shown in Table 4. Furthermore, the correlation coefficients R^2^ of the fitted curves of the modified Bingham model are all greater than 0.98. The goodness of fit indicates that the model has the good predictive capability.

### 3.2. Yield Stress

Yield stress refers to the corresponding shear stress, and whether the slurry begins to flow depends on the adhesion and friction between the particles in the slurry [36]. Figure 6 shows the relationship between the yield stress of basalt fiber-reinforced bentonite cement paste (BFBCP), the content, and the length of basalt fibers (BF). However, as basalt fiber (BF) content increases, the yield stress of the paste increases (Figure 6a). The yield stress of bentonite cement paste (BCP) without fiber is 16.94 Pa. Then, the yield stress of basalt fiber-reinforced bentonite cement paste (BFBCP) was approximately linear to the basalt fiber (BF) content in the range of 14.93–141.17% compared to that of bentonite cement paste (BCP). The basalt fibers (BF) are embedded between fine particles such as steel in the slurry. When the slurry flows to overcome the friction between particles caused by the incorporation of basalt fibers (BF), the yield stress of basalt fiber-reinforced bentonite cement paste (BFBCP) increases with the increase in fiber content. As the fiber length increases, the yield stress of the paste also increases (Figure 6b). The force is approximately exponential with the aspect ratio of the fiber. Therefore, samples increased by 200.19% from bentonite cement paste (BCP) compared with the yield stress of 0.6 basalt fibers (BF) and 12 bentonite cement paste (BCP). This occurs as the length of the fibers increases the area in contact with the particles and the friction between them and the slurry.

### 3.3. Plastic Viscosity

Plastic viscosity is a property that restricts flow due to the flocculation or reticulation of the internal structure of the slurry [37]. Figure 7 indicates the relationship between basalt fiber (BF) content and the length of plastic viscosity. The incorporation of basalt fibers (BF) can increase the plastic viscosity of basalt fiber-reinforced bentonite cement paste (BFBCP) (Figure 7a). The plastic viscosity of 0.8 basalt fibers (BF) and 9.0 bentonite cement paste (BCP) was 424.91 Pa·s, which makes it the highest compared to the other samples. This was resulted from the perspective of fiber agglomeration. Then, with an increase in fiber content, the probability of fiber-to-fiber contact is greater than that of fiber-to-slurry contact [38]. Thus, an increased fiber contact probability makes the fiber network structure more compact and reduces the slurry’s ability to flow freely.

The plastic viscosity of basalt fiber-reinforced bentonite cement paste (BFBCP) increases with an increase in fiber length linearly (Figure 7b). Moreover, when fiber length increases, it aggregates more and presents resistance to the free flow of slurry due to the dominant flow behavior of the slurry and increased plastic viscosity. In addition, when the fiber length and content increased from one level to another, the plastic viscosities of basalt fiber-reinforced bentonite cement paste (BFBCP) increased by 13.96~30.96% and 29.88~45.71%, respectively. Then, this shows that the effect of fiber content on plastic viscosity is greater than that of fiber length on plastic viscosities.

### 3.4. Unconfined Compressive Strength

Figure 8a illustrates the unconfined compressive strength (UCS) of basalt fiber-reinforced bentonite cement paste (BFBCP) with a given 9 mm length of basalt fibers (BF) (it means that BF9BCP) against basalt fiber (BF) content. Unconfined compressive strength (UCS) of basalt fiber-reinforced bentonite cement paste (BFBCP) increased at first and then decreased with basalt fiber (BF) content between 7 days and 28 days. The optimum basalt fiber (BF) content for 7 days and 28 days basalt fiber-reinforced bentonite cement paste (BFBCP) was 0.6%. The optimum unconfined compressive strength (UCS) for 7 days and 28 days toward BF9BCP was 3.45% and 19.17% higher than that of bentonite cement paste (BCP) (without basalt fibers (BF) inclusion). However, adding excessive basalt fibers (BF) may harm the unconfined compressive strength (UCS) gained. When the basalt fiber (BF) content is too high, they may entwine basalt fiber-reinforced bentonite cement paste (BFBCP) and disrupt the effectiveness of bonding between soil particles. This finding is in agreement with Chen [39]. In addition, fiber reinforcement ranged from 2.02% to 3.45% and 7.95% to 19.17% for the 7-day and 28-day care specimens, respectively. This indicates that the reinforcing effect of the fibers tends to increase with increasing curing time. As the time of curing increases, more cementing gels are produced within the basalt fiber-reinforced bentonite cement paste (BFBCP), which enhances the bond between the fibers and the matrix.

Figure 8b presents the unconfined compressive strength (UCS) of 0.6 basalt fiber-reinforced bentonite cement paste (BFBCP) versus basalt fiber (BF) length. However, unconfined compressive strength (UCS) increases and decreases with increasing basalt fiber (BF) content and similar to the effect of fiber content on unconfined compressive strength (UCS). The optimum basalt fiber (BF) length for 7-day and 28-day basalt fiber-reinforced bentonite cement paste (BFBCP) was 9 mm, resulting from the increase in fiber length that can increase the area between the fiber and the matrix. However, the unconfined compressive strength (UCS) of 0.6 basalt fibers (BF) and 12 bentonite cement paste (BCP) is reduced by 1.35% compared to that of 0.6 basalt fiber (BF) and 9 bentonite cement paste (BCP) due to the difficulty of dispersing of 12 mm from the fiber clusters into the fiber monofilaments under high-speed mixing. In addition, the contribution of fiber length to unconfined compressive strength (UCS) ranged from 16.99% to 19.96% was lower than fiber content. Thus, the effect of fiber length on unconfined compressive strength (UCS) is less than that of fiber content.

### 3.5. Splitting Tensile Strength

The effect of basalt fiber (BF) content on the splitting tensile strength (STS) (Figure 9a) was explored. However, the splitting tensile strength (STS) of 7-day and 28-day basalt fiber-reinforced bentonite cement paste (BFBCP) increased its peak and decreased with increasing basalt fiber (BF) content. Compared to bentonite cement paste (BCP) (without basalt fiber (BF)), splitting tensile strength (STS) increased by 9.78, 14.51, 28.21, and 19.16% for basalt fiber-reinforced bentonite cement paste (BFBCP) with a fiber content of 0.2, 0.4, 0.6, and 0.8%, respectively. The optimum BF content for 7 days and 28 days of basalt fiber-reinforced bentonite cement paste (BFBCP) was 0.4 and 0.6%, indicating that the optimum basalt fiber (BF) content tended to increase with curing age evolution. More gels and basalt fibers (BF) increased, whereas matrix cement structures formed as time increased continuously. Furthermore, the results for the optimum 7 days and 28 days of basalt fiber-reinforced bentonite cement paste (BFBCP) were, respectively, 20.41 and 28.21% higher than that of bentonite cement paste (BCP). The fiber network structure induced by basalt fiber (BF) and the interfacial friction between basalt fiber (BF) and matrix improved the tensile resistance of basalt fiber-reinforced bentonite cement paste (BFBCP). However, adding excessive basalt fibers (BF) will affect the gains in splitting tensile strength (STS) and be related to the difficulties of dispersing excessive amounts of fiber.

The effect of basalt fiber (BF) length on the splitting tensile strength (STS) of C0.9N0.6PDS is shown in Figure 9b. The splitting tensile strength (STS) of basalt fiber-reinforced bentonite cement paste (BFBCP) increased with the increase in basalt fiber (BF) length. Compared to bentonite cement paste (BCP) (without basalt fiber (BF)), splitting tensile strength (STS) increased by 9.67, 17.29, 28.21, and 20.49% for basalt fiber-reinforced bentonite cement paste (BFBCP) with a fiber content of 0.2, 0.4, 0.6, and 0.8%, respectively. Therefore, the basalt fiber (BF) length has a significant effect on the splitting tensile strength (STS) development of basalt fiber-reinforced bentonite cement paste (BFBCP) due to the increase in fiber length that enhances the bridging effect of the fiber and encourages the fiber to form a wider force transmission system within the matrix [40]. As a result, the specimen can be more uniformly loaded internally, thereby increasing the tensile strength of basalt fiber-reinforced bentonite cement paste (BFBCP). However, the splitting tensile strength (STS) of 0.6 basalt fiber (BF) and 12 bentonite cement paste (BCP) is less than that of basalt fiber-reinforced bentonite cement paste (BFBCP) of 0.6 basalt fibers (BF) and 12 bentonite cement paste (BCP). It may be attributed to the long basalt fibers (BF) interwoven inside the basalt fiber-reinforced bentonite cement paste (BFBCP) that formed the agglomeration.

### 3.6. Microstructure

#### 3.6.1. The Fiber Reinforcement Mechanism

Micromechanical mechanisms, such as fiber filling, fiber bridging, and fiber pull out, can be used to explain the reinforcing effect of basalt fibers. Basalt fiber can be used as microaggregates to increase the mechanical strength of basalt fiber-reinforced bentonite cement paste (BFBCP) because it is a mineral substance. The addition of fibers contributed to a denser and more homogeneous structure (Figure 10a). Fiber bridging reinforced mechanism for fiber-reinforced materials during the destruction process. Bridging fiber can reduce crack development and bear loads; this has been the case in basalt fiber-reinforced bentonite cement paste (BFBCP) (Figure 10b). Basalt fibers can pull out and absorb energy through the friction between fibers and matrix (Figure 10c,d). Therefore, friction depends on the strength of the bond between the fiber and the bonding particles. It is well known that the main hydration products (C-S-H gels, AFt, etc.) can improve the strength performance by cementing fine grains and filling pore spaces [41]. As shown in Figure 10d, the number of fine fibrous CSH gels and rod-shaped AFt crystals was distributed on the fiber surface and the pull-out traces resulting in a limitation of the relative movement between the basalt fiber and the matrix.

The energy-dispersive x-ray spectroscopy (EDS) patterns with CSH gels and Aft (Figure 11a,b) support the scanning electron microscopy (SEM) analysis on basalt fiber-reinforced bentonite cement paste (BFBCP). The rod-like structure (Figure 11a) is classified as Aft because it has a lot of sulfur, silicon, calcium, and oxygen content, while the gel-like structure (Figure 11b) is known as a CSH due to its higher oxygen content and calcium-silica ratio. The findings are in agreement with Wang et al. [42].

#### 3.6.2. Fiber Microscopic Distribution

Uniform fiber distribution is a prerequisite for good mechanical properties in cementitious composites. Figure 12 shows the microstructure of 0.6 basalt fibers (BF) in 9 bentonite cement paste (BCP) specimens and 0.6 basalt fibers (BF) in 12 bentonite cement paste (BCP) specimens. Most fibers are monofilaments and distributed uniformly (Figure 12a), whereas others are distributed in fiber clusters (Figure 12b).

As shown in Figure 8, the strength of the 0.6 basalt fiber (BF) 9 bentonite cement paste (BCP) specimen is higher compared to that of 0.6 basalt fiber (BF) 12 bentonite cement paste (BCP). Once the fibers form bundles or clusters, their strengthening efficiency decreases sharply. Despite the fiber content and length increase, the rheological properties of the paste decrease, making the slurry hard to mix with the uneven distribution of fiber and increasing the number of fiber-to-fiber interfaces within the matrix. Then, the absence of cementation of hydration products in this area makes it a weak structural surface within the matrix [43].

## 4. Conclusions

In this study, basalt fiber (BF) content and length influenced the rheological properties and mechanical strength of basalt fiber-reinforced bentonite cement paste (BFBCP) and the relationships between them. The following are the conclusions drawn from the results presented in this study:(1)The rheological properties of basalt fiber-reinforced bentonite cement paste (BFBCP) conform to the modified Bingham rheological model. An increase in fiber content and length did not change the bentonite cement paste (BCP) rheological model. The yield stress and plastic viscosity of basalt fiber-reinforced bentonite cement paste (BFBCP) increased with the increase in content and length of the fiber. When the basalt fiber (BF) content rises from 0% to 0.8%, the yield stress and plastic viscosity of the basalt fiber-reinforced bentonite cement paste (BFBCP) increase by 123.84% and 162.14%, respectively, because the fine rod-like fiber forming a net-like structure in the slurry increases the frictional resistance between the slurry particles.(2)The unconfined compressive strength (UCS) of the basalt fiber-reinforced bentonite cement paste (BFBCP) increased before basalt fiber (BF) content and length reached a certain level, and then started to decrease. The addition of 0.6% of basalt fiber (BF) content resulted in the highest unconfined compressive strength (UCS) value (3.416 MPa) and an increase of 17.19% when compared to bentonite cement paste (BCP). When the fiber length increases from 3 mm to 12 mm, the growth rate strength of basalt fiber-reinforced bentonite cement paste (BFBCP) increases first and then decreases; the growth rate ranges from 16.6% to 19.9%. The contribution of fiber content to unconfined compressive strength (UCS) is greater than that of fiber length.(3)The splitting tensile strength (STS) of the basalt fiber-reinforced bentonite cement paste (BFBCP) increased before the basalt fiber (BF) content and the length decreased at a certain level. The optimum fiber content is affected by curing time. The optimum fiber content of the sample cured for 7 days and 28 days is 0.4% and 0.6%, respectively. Moreover, the optimal growth rate of splitting tensile strength (STS) for basalt fiber-reinforced bentonite cement paste (BFBCP) in 28 days of curing was 28.21%, compared to 0.6% for bentonite cement paste (BCP). Then, the fiber’s tensile strength contributes more than its compressive strength.(4)According to microstructure analysis, the incorporation of basalt fibers (BF) is important to obtaining a well-interconnected skeletal structure of basalt fiber-reinforced bentonite cement paste (BFBCP). Furthermore, the cement hydration products improved adhesion to the matrix and increased the energy required for fiber pull out. When the specimen was subjected to impact loading, excess or overlong basalt fibers within the basalt fiber-reinforced bentonite cement paste (BFBCP) formed a weak surface, resulting in sliding failure at once due to the large number of fiber-to-fiber interfaces from the non-uniform distribution of the fibers.

## Figures and Tables

**Figure 1 materials-16-03226-f001:**
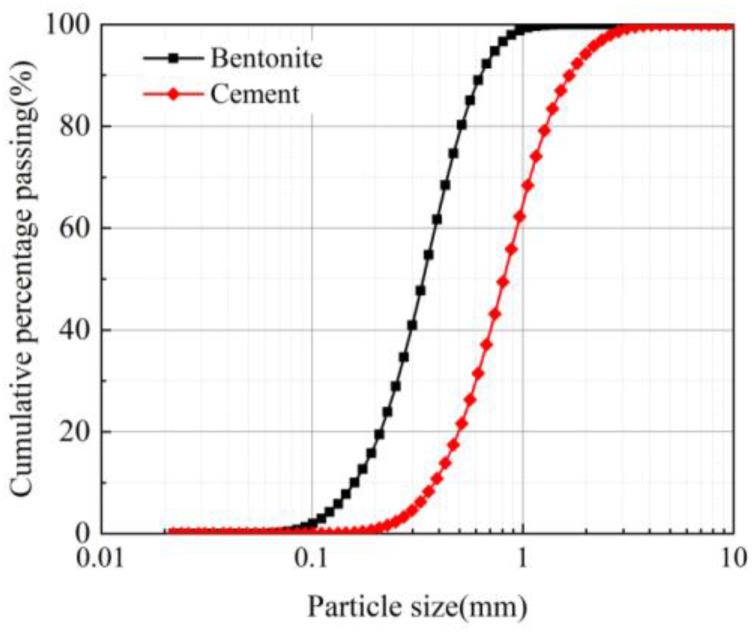
Gradation curve of bentonite and cement.

**Figure 2 materials-16-03226-f002:**
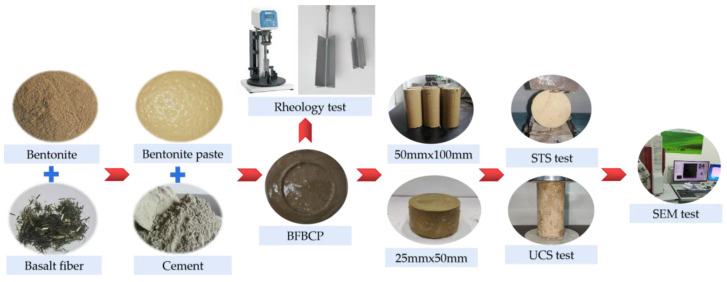
Schematic diagram of sample preparation and testing.

**Figure 3 materials-16-03226-f003:**
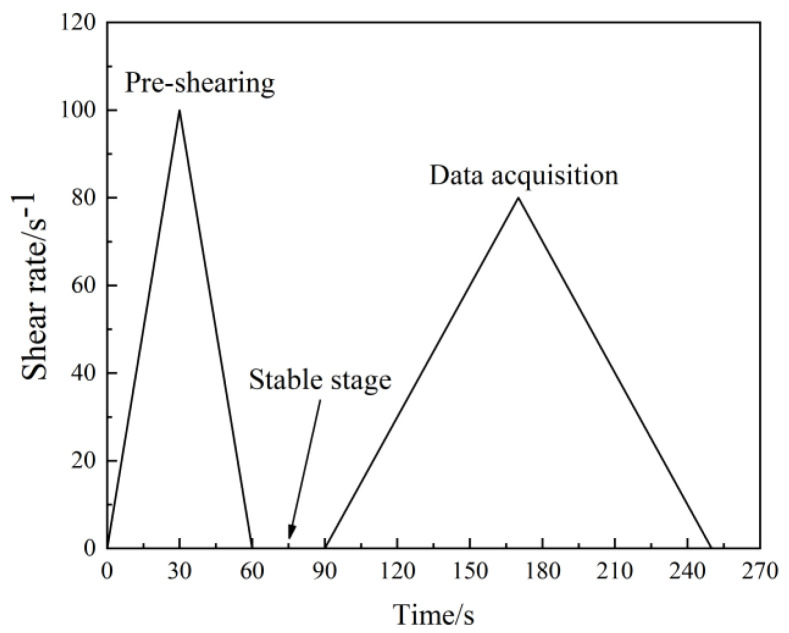
Test procedures of the rheological instrument.

**Figure 4 materials-16-03226-f004:**
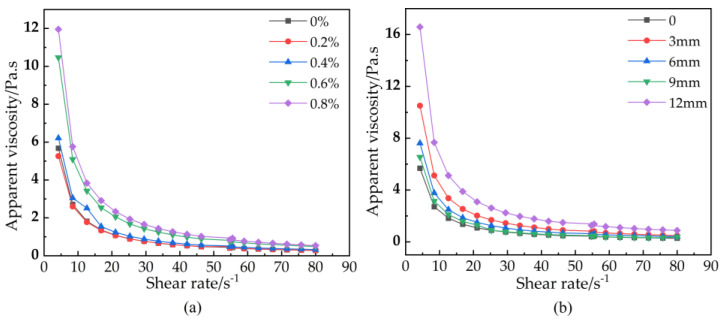
The relationship between shear rate and apparent viscosity: (**a**) fiber content, (**b**) fiber length.

**Figure 5 materials-16-03226-f005:**
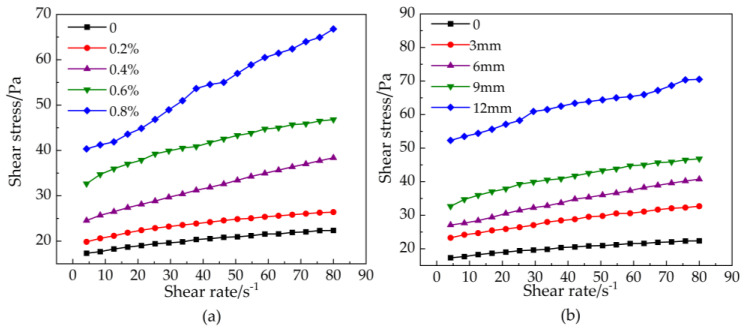
The relationship between shear rate and shear stress: (**a**) fiber content; (**b**) fiber length.

**Figure 6 materials-16-03226-f006:**
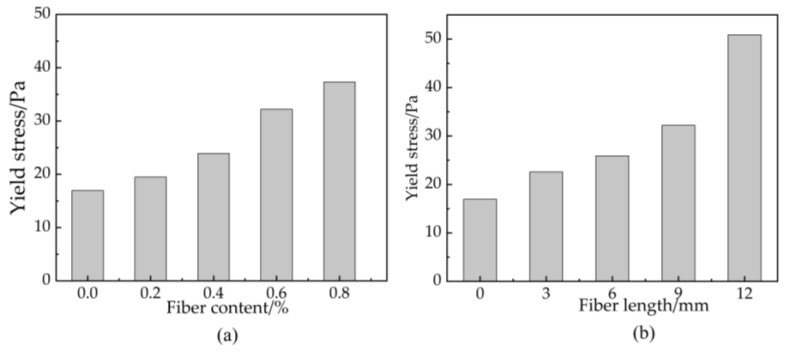
Effect of basalt fiber on the yield stress of BFBCP: (**a**) fiber content, (**b**) fiber length.

**Figure 7 materials-16-03226-f007:**
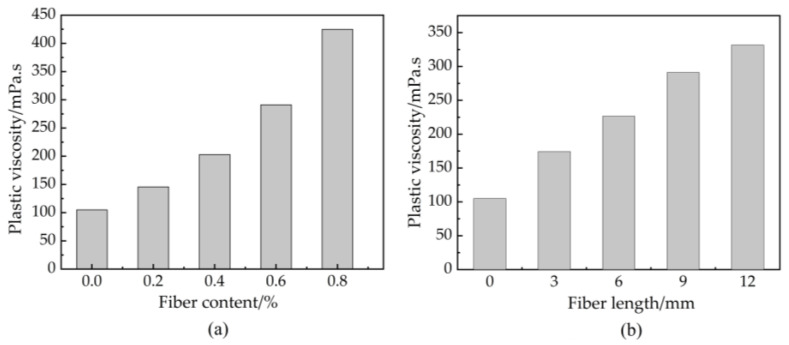
Effect of basalt fiber on plastic viscosity of BFBCP: (**a**) fiber content, (**b**) fiber length.

**Figure 8 materials-16-03226-f008:**
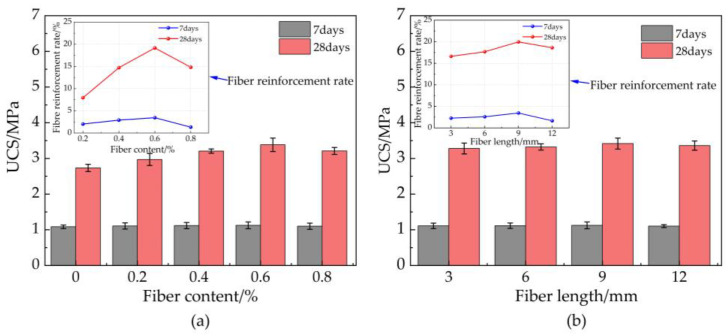
Effect of basalt fiber on UCS of BFBCP: (**a**) fiber content; (**b**) fiber length.

**Figure 9 materials-16-03226-f009:**
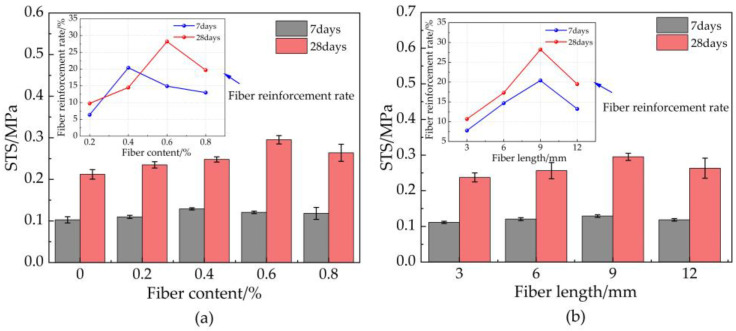
Effect of basalt fiber on STS of BFBCP: (**a**) fiber content; (**b**) fiber length.

**Figure 10 materials-16-03226-f010:**
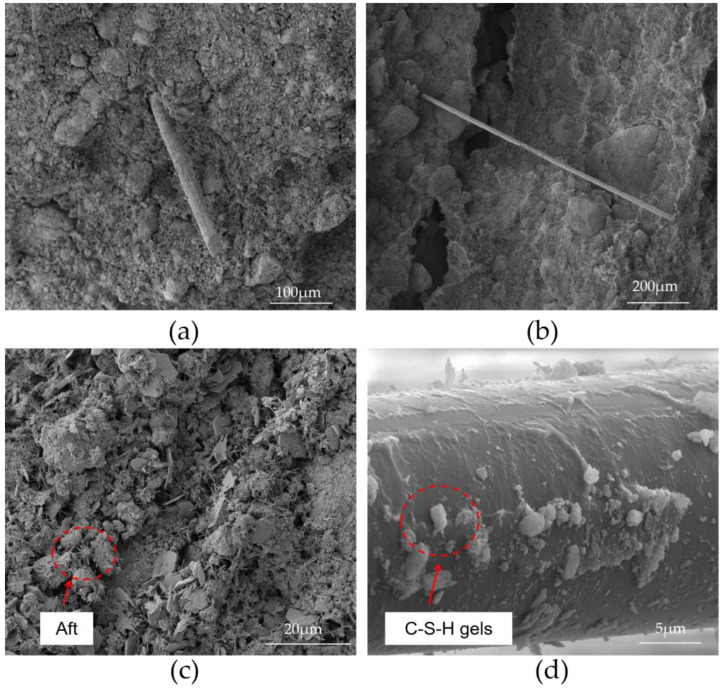
SEM images of typical specimens: (**a**) typical cross-section of a BFBCP; (**b**) fiber bridging; (**c**) Aft; (**d**) C-S-H gels.

**Figure 11 materials-16-03226-f011:**
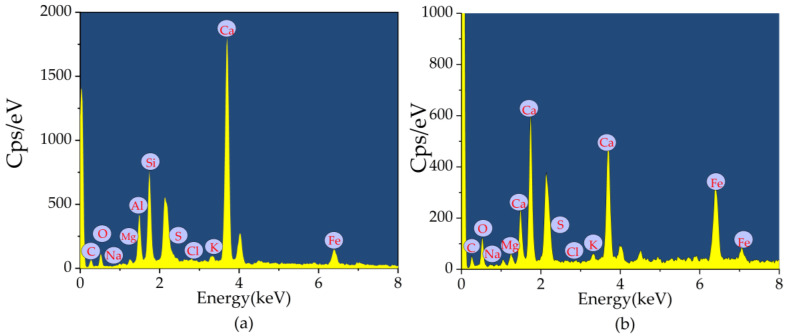
EDS patterns of Figure 9a,b: (**a**) Aft; (**b**) C-S-H.

**Figure 12 materials-16-03226-f012:**
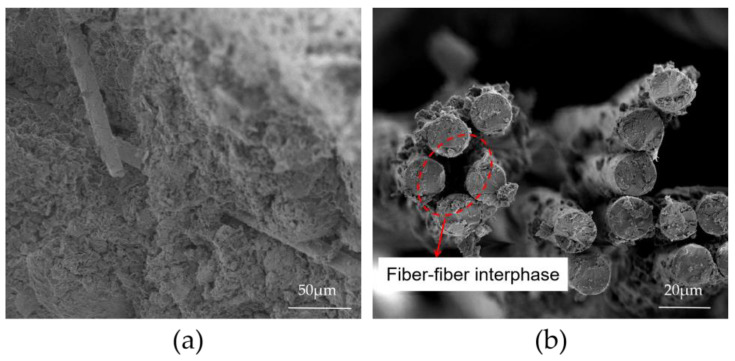
SEM images of different BFBCP specimens: (**a**) 0.6 BF9BCP; (**b**) 0.6 BF12BCP.

**Table 1 materials-16-03226-t001:** The major chemical composition of bentonite and cement as wt%.

Chemical Composition	CaO	SiO_2_	Al_2_O_3_	Fe_2_O_3_	MgO	Ti_2_O_3_	SO_3_	Na_2_O	Loss on Ignition
Bentonite	0.54	63.78	14.42	3.63	1.71	0.51	-	0.97	9.91
Cement	51.42	24.99	8.26	4.03	3.71	-	2.51	-	3.31

**Table 2 materials-16-03226-t002:** Physical and mechanical properties of basalt fiber.

Fiber Diameter /μm	Fiber Type	Average Length (mm)	Density /kg·m^3^	Elongation at Break/%	Elastic Modulus/GPa	Tensile Strength/MPa
13–17	Short cut	3, 6, 9, 12	2650	3.1	75	1050

**Table 3 materials-16-03226-t003:** Mixture design of test materials.

Mix ID	W/C	Cement (wt%)	Bentonite (wt%)	BF Content (vf%)	BF Length (mm)	Curing Time (Days)
BCP	0.8	40	60	0	-	7, 28
0.2BF9BCP	0.8	40	60	0.2	9	7, 28
0.4BF9BCP	0.8	40	60	0.4	9	7, 28
0.6BF9BCP	0.8	40	60	0.6	9	7, 28
0.8BF9BCP	0.8	40	60	0.8	9	7, 28
0.6BF3BCP	0.8	40	60	0.6	3	7, 28
0.6BF6BCP	0.8	40	60	0.6	6	7, 28
0.6BF9BCP	0.8	40	60	0.6	9	7, 28
0.6BF12BCP	0.8	40	60	0.6	12	7, 28

**Table 4 materials-16-03226-t004:** Fitting analysis of the modified Bingham rheological model.

Mix ID	Fitting Equation	Yield Stress/Pa	Plastic Viscosity/mPa·s	R^2^_Adj_
BCP	$\tau =16.948+0.104\dot{\gamma }-4.683{\dot{\gamma }}^{2}$	16.948	104.984	0.996
0.2BF9BCP	$\tau =19.475+0.145\dot{\gamma }-7.561{\dot{\gamma }}^{2}$	19.475	145.723	0.996
0.4BF9BCP	$\tau =23.905+0.202\dot{\gamma }-2.722{\dot{\gamma }}^{2}$	23.905	202.982	0.999
0.6BF9BCP	$\tau =32.215+0.291\dot{\gamma }-0.001{\dot{\gamma }}^{2}$	32.215	291.169	0.994
0.8BF9BCP	$\tau =37.285+0.424\dot{\gamma }-7.420{\dot{\gamma }}^{2}$	37.285	424.524	0.993
0.6BF3BCP	$\tau =22.597+0.174\dot{\gamma }-6.047{\dot{\gamma }}^{2}$	22.597	174.346	0.998
0.6BF6BCP	$\tau =25.890+0.226\dot{\gamma }-5.040{\dot{\gamma }}^{2}$	25.890	226.545	0.988
0.6BF9BCP	$\tau =32.215+0.291\dot{\gamma }-0.011{\dot{\gamma }}^{2}$	32.215	291.169	0.994
0.6BF12BCP	$\tau =50.872+0.331\dot{\gamma }-0.030{\dot{\gamma }}^{2}$	50.872	331.652	0.984

## Data Availability

The data presented in this study are available on request from the corresponding author. The data are not publicly available due to the data also forms part of an ongoing study.

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
