# Peer review of "Rheological and Mechanical Properties of Bentonite–Cement Paste Reinforced with Basalt Fibers"

_materials, 2023, doi:10.3390/ma16083226_

Round 1

Reviewer 1 Report

Dear authors, 
I attach a pdf with some minor comments.

Author Response

Response to Reviewer 1 Comments

Point 1: shouldn't you start with table 1?

Response 1: Thank you for your advice, the correct formulation of Table 1 has been added。

Point 2: do you have any std dev, error, uncertainty for the experimental results? (all of them). or is it single tests?

Response 2: Thank you for your advice, the yield stress and plastic viscosity are obtained from calculations of the rheological model. This is a single test, so no allowance is made for error.

Point 3: I would try to link the coding names from the above table。

Response 3: Thank you for your advice, The tables in the manuscript have been correctly cod.

Reviewer 2 Report

Dear Authors,

Thank you for elaborating the research results on the topic of rheological and nonchanical properties of Bentonite-Cement Paste reinforced with Basalt Fibers.

In terms of inclusion, it is suitable for the selected journal.

The overall impression of the article is positive - the number of tests, the approach, the formality, and the evaluation are of a high scientific standard.

I have a few remarks and comments. Here are the evaluations for each part of the text:

The title and abstract are clear. Right.

The introduction needs to be handled a little better - first of all, abbreviations need to be restated and explained, even if they are in the abstract - the first use of an abbreviation needs the whole term.

Further, the overall assessment of the rationale for your research in the context of civil engineering is lacking - I would recommend mentioning other materials suitable as alternatives for tunnel reinforcement, other alternatives for fibres, etc.

For example:

10.3390/buildings11100489

10.3390/buildings12081082

In section 2.1 Materials and Mix Design you state " And it is known through the preliminary exploration experiments...." you must provide a reference to the publication or other source of information from which you have drawn. 

I would expect the chemical composition of each base material first and then a description of the mixtures.

References to test procedures and standards are insufficient. The description is too brief.

I would expect you to evaluate the short term mechanical properties first and then the rheological properties - as they take longer to evaluate.

The description of the results is fine.

On the other hand, the conclusions are rather repetitive results - the conclusions need to be more clearly defined in scientific and practical terms. 

Reviewer 3 Report

The presented article indicates the possibility of using basalt fiber in the composition of cement. Materials and methods are standard and do not raise questions. The scheme of sample preparation and testing is shown. The obtained conclusions are confirmed by experiments.

I think this is a great article.

Author Response

Point 1: The presented article indicates the possibility of using basalt fiber in the composition of cement. Materials and methods are standard and do not raise questions. The scheme of sample preparation and testing is shown. The obtained conclusions are confirmed by experiments.

I think this is a great article.

Response 1: We deeply appreciate the time and effort you’ve spent in reviewing our manuscript. Your review is very constructive and detailed. Thank you very much for your recognition of our work.

Reviewer 4 Report

In this work, the authors have investigated the rheological and mechanical properties of BF reinforced cement paste by various mechanical methods. 

Then, They used the Binghan rheolgogical model to curve-fit the rheological data and to obtain the yield stress. However, the results are not impressive and the results are predictable. 

- First of all, there are so many clumsy expression and hard to understand. Many wrong grammars are found. 

- Show the fitted curve with the Bingham model in Fig. 5 to verify the curve fitting results. Since the modified Bingham model in eq. 2 is just a parabolic curve that does not fitt well with the experimental results shown in Fig. 5. 

- I strongly suggest them to investigate the interfacial bonding strength between the fibers and cement to in-depth understand the mechanical and rheological behaviors. 

At this current form, I do not find any good reason for this manuscript to be accepted. 

Round 2

Reviewer 2 Report

Thanks, no it is ok.

Author Response

We deeply appreciate the time and effort you’ve spent in reviewing our manuscript. Your review is very constructive and detailed. Thank you very much for your recognition of our work.

Reviewer 4 Report

Although I am still doubt about the novelty and interest of this work, the authours have addressed and revised the manuscript in a good shape. 

Author Response

We thank you very much for your recognition of our work. Also, we sincerely appreciate your comments for helping us to improve our manuscript. Your review is very constructive and detailed. In future research, we will further improve the novelty and interest of our research.